# The Tip of *Brucella* O-Polysaccharide Is a Potent Epitope in Response to Brucellosis Infection and Enables Short Synthetic Antigens to Be Superior Diagnostic Reagents

**DOI:** 10.3390/microorganisms10040708

**Published:** 2022-03-25

**Authors:** Lucy Duncombe, Laurence Howells, Anna Haughey, Andrew V. Taylor, Daryan Kaveh, Sevil Erdenliğ Gϋrbilek, Anne Dell, Paul G. Hitchen, Stuart M. Haslam, Satadru Sekhar Mandal, N. Vijaya Ganesh, David R. Bundle, John McGiven

**Affiliations:** 1Department of Bacteriology, Animal & Plant Health Agency, OIE/FAO Brucellosis Reference Laboratory, Woodham Lane, Addlestone, Surrey KT15 3NB, UK; lucy.duncombe@apha.gov.uk (L.D.); laurence.howells@apha.gov.uk (L.H.); anna.haughey@apha.gov.uk (A.H.); andrew.v.taylor@apha.gov.uk (A.V.T.); daryan.kaveh@apha.gov.uk (D.K.); 2Microbiology Department, Faculty of Veterinary Medicine, Harran University, 63500 Şanliurfa, Turkey; sevilerdenlig@yahoo.com; 3Department of Life Sciences, Imperial College London, London SW7 5NH, UK; a.dell@imperial.ac.uk (A.D.); p.hitchen@imperial.ac.uk (P.G.H.); s.haslam@imperial.ac.uk (S.M.H.); 4Department of Chemistry, University of Alberta, Edmonton, AB T6G 2G2, Canada; satadrusekhar@gmail.com (S.S.M.); nvijayaganesh@gmail.com (N.V.G.); dave.bundle@ualberta.ca (D.R.B.)

**Keywords:** *Brucella*, O-polysaccharide, epitope

## Abstract

Brucellosis is a global disease and the world’s most prevalent zoonosis. All cases in livestock and most cases in humans are caused by members of the genus *Brucella* that possess a surface O-polysaccharide (OPS) comprised of a rare monosaccharide 4-deoxy-4-formamido-D-mannopyranose assembled with α1,2 and α1,3 linkages. The OPS of the bacterium is the basis for serodiagnostic tests for brucellosis. Bacteria that also contain the same rare monosaccharide can induce antibodies that cross-react in serological tests. In previous work we established that synthetic oligosaccharides, representing elements of the *Brucella* A and M polysaccharide structures, were excellent antigens to explore the antibody response in the context of infection, immunisation and cross reaction. These studies suggested the existence of antibodies that are specific to the tip of the *Brucella* OPS. Sera from naturally and experimentally *Brucella abortus*-infected cattle as well as from cattle experimentally infected with the cross-reactive bacterium *Yersinia enterocolitica* O:9 and field sera that cross react in conventional serological assays were studied here with an expanded panel of synthetic antigens. The addition of chemical features to synthetic antigens that block antibody binding to the tip of the OPS dramatically reduced their polyclonal antibody binding capability providing conclusive evidence that the OPS tip (non-reducing end) is a potent epitope. Selected short oligosaccharides, including those that were exclusively α1,2 linked, also demonstrated superior specificity when evaluated with cross reactive sera compared to native smooth lipopolysaccharide (sLPS) antigen and capped native OPS. This surprising discovery suggests that the OPS tip epitope, even though common to both *Brucella* and *Y. enterocolitica* O:9, has more specific diagnostic properties than the linear portion of the native antigens. This finding opens the way to the development of improved serological tests for brucellosis.

## 1. Introduction

Serodiagnostic methods for brucellosis can be undertaken via a variety of platforms and using antigen preparations of variable composition [1]. What remains constant is that the critical serodiagnostic antigen for effective detection of antibodies raised by infection with *B. melitensis*, *B. abortus* and *B. suis* has always been the O-polysaccharide (OPS). This molecule is present in abundant amounts in all of the most sensitive and specific diagnostic antigens. The location and abundance of the OPS on the surface of these classical smooth species of *Brucella* [2,3], its repeating structure and potential ability to induce both T-independent and T-dependent antibody responses, by virtue of B-cell opsonisation of whole bacteria [4], contributes to its dominance of the host’s antibody response.

The OPS from *B. melitensis*, *B. abortus* and *B. suis* is relatively simple [5]. Other than the primer and adapter sugars it is an unbranched homopolymer of D-Rha4NFo sugars that are primarily α1,2 linked but with a variable ratio of α1,3 links, with the exception of *B. suis* biovar 2, which is exclusively α1,2 linked [6]. The position and abundance of these linkage types forms the basis of the A or M dominance of the OPS and so the strain [5]. The structure has been described as a block copolymer of two distinct sequences in which the reducing end polysaccharide of α1,2 linked residues constitutes the A antigen and the non-reducing end sequence consists of one or several tetrasaccharide sequences containing a single central α1,3 linkage which creates the M antigen [7,8]:[-2)Rha4NFo(α1-2)Rha4NFo(α1-3)Rha4NFo(α1-2)Rha4NFo(α1-]_m-_[-2)Rha4NFo(α1-]_n_

This represents the short form nomenclature for the *Brucella* OPS structure. A tetrasaccharide in which an α1,3 linkage occurs between the second and third Rha4NFo monosaccharides and creates the terminal M polysaccharide, where 2 or more of these tetrasaccharides are linked together via α1,2 linkages (m > 2). This polysaccharide is elaborated on an exclusively α1,2 linked polymer of Rha4NFo monosaccharides which constitute the A polysaccharide.

The similarity and coexistence of these polymers is diagnostically advantageous as a single OPS type from one strain makes a highly effective antigen for the detection of all field infections with these species [9,10]. Nonetheless, the immunological significance of the different structures is demonstrated by the variety of monoclonal antibodies (MAb) which identify different antibody epitopes, A, M and C/Y formed by the presence and position of linkage types [11]. A ‘C specific’ epitope has also been proposed but not defined although it must incorporate the same elements that make up those already described [12] as the OPS structure has been well defined and appears complete. These epitopes are conjoined and overlap within the native OPS.

Only relatively recently have the individual properties of these different epitopes been investigated and exploited for serological purposes [13], providing new diagnostics and fresh insights into vaccine design [14]. This has been achieved with synthetic D-Rha4NFo oligosaccharides and their conjugation to antigenically silent carriers that enable adsorption to immunoassay surfaces. Those with α1,3 links and minimal α1,2 links demonstrated superior capability to differentiate between antibodies raised by *Brucella* and *Brucella* OPS reactive antibodies raised by other Gram-negative bacteria, specifically *Yersinia enterocolitica* O:9 which possesses an exclusively α1,2 linked D-Rha4NFo homopolymer [13]. This organism is believed to be a significant source of false positive serological reactions [15,16,17] due to the similarity of its OPS structure to that of *Brucella* [18]. Despite the improved diagnostic specificity of these synthetic antigens, some cross-reaction against antibodies raised by infection with *Y. enterocolitica* O:9 remained. An explanation for this residual cross-reaction is the possible existence of a specific antibody epitope formed by the terminal (non-reducing) D-Rha4NFo, the tip of the OPS and a structure common between *Brucella* and *Y. enterocolitica* O:9.

Epitopes specific to the tip of the OPS have been described previously for other Gram-negative bacteria. The most well-studied is found in *Vibrio cholera* O:1 [19] where only the tip epitopes differentiate the Ogawa and Inaba serotypes. Both types produce substantial amounts of tip-specific antibodies [20]. In particular, the Inaba type OPS shows much structural similarity to *Brucella* OPS. A terminal OPS epitope has also been identified for *Francisella tularensis* [21], *Burkholderia pseudomallei* and *Burkholderia mallei* [22].

Previous investigation of the polyclonal anti-*Brucella* OPS antibody response provides evidence that it also comprises antibodies that are specific to the terminal structure [14]. The terminal D-Rha4NFo is unique within the *Brucella* OPS as it is the only unit to possess hydroxyl groups on both the second and third carbons. In all other cases within the polymer one or other of these carbons is part of the glycosidic linkage. This terminal structure is present in the OPS of *B. melitensis*, *B. abortus, B. suis*, *Y. enterocolitica* O:9 and the synthetic oligosaccharides previously evaluated for serodiagnosis [13]. An oligo- or polysaccharide that is capped, so that this terminal feature is not present, would possess only the linear epitope. This capping can be achieved by attachment through the non-reducing terminus, or by introduction of substituents to synthetic oligosaccharides that prevent the formation of antibody binding site hydrogen bonds to hydroxyls at either or both 2,3 positions of Rha4NFo. Alternatively, complementarity with the antibody binding site may also be abrogated by substitution of a terminal Rha4NFo residue by mannose. This retains monosaccharide stereochemistry but abolishes a formamido group and substitutes a bulky hydrophilic C-5 hydroxylmethyl group for a hydrophobic methyl group (in subsequent text we also refer to this substitution as capping with mannose).

The aim of this study was to identify the impact of the terminal structure on the detection of antibodies reactive to the *Brucella* OPS. Serum from cattle was probed with a panel of native and synthetic antigens expanded to include those capping the putative tip epitope. This serum included those from cattle experimentally infected with either *B. abortus* or *Y. enterocolitica* O:9, field sera *from B. abortus* infected cattle, false positive serological reactors, and sera from randomly sampled non-infected cattle. These defined antigens have identified a prominent tip epitope in the polyclonal responses of cattle to the immunodominant cell wall lipopolysaccharide (LPS) of these organisms that has previously been masked by the use of OPS as a diagnostic tool. Despite its relatively simple structure, *Brucella* OPS presents several distinct epitopes and during the course of natural infection the variable magnitude of polyclonal responses to each of these can cloud the diagnostic identification of infected animals. The results verified the existence of the tip epitope, demonstrated its substantial impact on antibody binding and identified improved diagnostic antigens for detection of *Brucella* infection.

## 2. Materials and Methods

### 2.1. Origin of Bacterial Antigens

Native smooth lipopolysaccharide (sLPS) and OPS antigens were extracted from *B. abortus* strain S99, *B. melitensis* strain 16M and *B. suis* strain Thomsen (biovar 2). The strains were grown on serum dextrose agar plates at 37 °C at 10% CO_2_ for 3–5 days until late-log phase and harvested into phosphate buffered saline (PBS). *Y. enterocolitica* serotype O:9 (biotype 2), from which OPS was extracted, was grown on nutrient agar plates at 25 °C, atmospheric CO_2_ levels for 3 days until late-log phase and then harvested into PBS. All harvested cells were incubated at 80 °C until confirmed non-viable by culture prior to further manipulation.

### 2.2. Extraction and Preparation of Native Antigens

sLPS was prepared from *B. abortus*, *B. melitensis*, *B. suis* and *Y. enterocolitica* O:9 by hot-phenol extraction [9,23]. The OPS was extracted from sLPS by mild acid hydrolysis [5]. Following hydrolysis, the precipitated lipid A was removed as the pellet following centrifugation at 17,000× *g* for 30 min. The supernatant was buffer exchanged into water by size exclusion chromatography using Sephadex G-25 which also removed low molecular weight impurities including cleaved core antigens. The purity of the sLPS and OPS was confirmed using SDS-PAGE silver staining, Bradford assay (Pierce™ Coomassie Protein Assay Kit, Fisher Scientific UK Ltd., Loughborough, UK), endotoxin assay (PyroGene™ Recombinant Factor C Assay, Lonza, Basel, Switzerland), UV 210–280 nm HPLC and mass spectrometry. The OPS was chemically modified (to produce mOPS) by oxidation of the terminal D-Rha4NFo and in so doing not only eliminating the putative tip epitope, but also generating chemically reactive groups at this site for onward conjugation to functionalised ELISA plates. Briefly, the OPS, at 2 mg/mL, was oxidised by incubation in 10 mM sodium metaperiodate in 50 mM sodium acetate buffer at pH 5.5 at 4 °C for 1hr in the dark [14].

### 2.3. Synthetic Antigens

Synthetic oligosaccharide BSA conjugates were prepared as described previously [14,24,25,26]. Quantification of loading of the synthetic oligosaccharides to bovine serum albumin (BSA) was performed by MALDI-ToF and HPLC-ESI-MS. These conjugates are listed as (1 to 21) in Table 1 which gives the full chemical notation, the type of linker used for conjugation to the BSA (dibutyl squarate [Sq] or disuccinimidal glutarate [DSG]) the average number of oligosaccharides conjugated per BSA and the primary reference for each conjugate as well as the shorthand notation that will be used within this text. Full chemical structures for each oligosaccharide plus linker are shown in Appendix A.

### 2.4. Serum Samples

For the initial evaluation of the impact of the presence or absence of the putative tip epitope, a panel of 12 sera from individual *B. abortus* field infected cattle and 4 sera from non-*Brucella* infected cattle were evaluated (shown as ‘Infect’ 1–12 and ‘Ave Non-infect’ in Figures 1 and 2). This panel, with 12 ‘infect’ sera, consisted of 6 serum samples 1 to 6 from individual animals derived from Great Britain, each confirmed by bacterial culture [27] from fresh tissues to be naturally infected with A dominant strains of *B. abortus* (biovars 1 or 3). The remaining 6 ‘infect’ serum samples, 7 to 12, in this panel were derived from cattle in Turkey and were serologically positive by RBT and CFT. These samples were from a herd where at least one animal had been confirmed as being naturally infected with an A dominant strain of *B. abortus* (biovars 1 or 3) by culture from milk samples. Another four samples were evaluated which were taken after 2007 (long after the last recoded case of brucellosis in the UK) from non-*Brucella* infected cattle from within Great Britain (GB), the average of these four samples was calculated and shown as ‘non-infect’ on Figures 1 and 2. To evaluate differences in the titre of antibody binding between the capped and non-capped oligosaccharides, six of these sera from the 12 infected cattle and two of the sera from the non-infected cattle were titrated against the antigens (Appendix A).

A second panel of serum samples from eight cattle, experimentally infected as described previously [28], was also evaluated. This serum panel consisted of four cattle infected with *B. abortus* strain 544 (an A-dominant strain) and four with *Y. enterocolitica* O:9. Samples that were taken 3, 7, 16, 24 and 53 weeks post infection from each animal (*n* = 8) at each time point were evaluated with a panel of diagnostic antigens. All experimental animal procedures were conducted in accordance with the United Kingdom Animal (Scientific Procedures) Act of 1986.

A third serum panel of 321 serum samples was evaluated with a selection of antigens. This panel consisted of 42 sera from *B. abortus* field infected cattle (derived from Turkey where *Brucella* was isolated from at least one animal in the herd and also from an archive of sera from cattle in Great Britain where *B. abortus* had been isolated from each animal). Also tested in this serum panel were sera from 39 animals from GB collected after declaration of officially brucellosis free (OBF) status but which were identified as false positive reactors by at least one conventional assay (SAT, CFT, RBT or iELISA), henceforth described as False Positive Serological Reactor sera (FPSRs). Epidemiological investigations at the time revealed no other evidence of *Brucella* infection. Infection of field (and experimental) cases was confirmed by the isolation of *Brucella* by bacterial culture [27] from either tissue (lymph nodes, blood, or abortion material), vaginal swabs, or milk culture, depending upon the particular case. A further 240 sera from randomly sampled non-*Brucella* infected cattle (from GB) were also evaluated in the third serum panel. No animals were sampled more than once.

Sera derived from mice immunised with a *B. abortus* S99 OPS–tetanus toxoid glycoconjugate were also further evaluated in this study [26].

### 2.5. Serological Methods

The iELISAs performed with the cattle sera and the BSA conjugated synthetic oligosaccharide antigens were conducted as described previously [13]. The iELISA with the sLPS (derived from *B. abortus* S99) was conducted using the same method with two minor adaptations. The sLPS was coated to the ELISA plate at a concentration of 0.5 µg/mL and the conjugate (HRP conjugated rabbit anti-bovine IgG) was used at a 1/4000 concentration (compared to 1/2000 for the synthetic oligosaccharide conjugates). The iELISA with the mouse serum was conducted as described previously [26]. The iELISA with BM40 MAb was conducted using plates prepared with BSA conjugated synthetic oligosaccharide antigens using the method as for serum. The BM40 MAb was added to the wells of the plates in 100 µL volumes of PBS at the final concentrations shown in Appendix A (from 20 µg/mL to 0.04 µg/mL in doubling dilutions). Plates were incubated at room temperature (RT) for 1 h on a rotary shaker at 120 rpm, then washed four times with 200 µL/well PBST and tapped dry. Rabbit anti-mouse IgG: HRP (Dako) was diluted at 1.3 µg/mL in PBST and added to the plates, 100 µL/well. The plates were incubated at RT for 1 h, on a rotary shaker at 120 rpm. Then washed and dried as previously. Plates were read as described for serum iELISA. The results were expressed as a percentage of a common positive control sample (BM40 at 5 µg/mL) and antigen (*B. melitensis* 16 M sLPS at 0.5 µg/mL) combination that was used on each plate.

The oxidised (m)OPS was diluted to 0.125 µg/mL in 0.1 M sodium acetate buffer pH 5.5, mixed, and 100 µL of this was added to each well of a CarboBIND^TM^ ELISA plate (Corning, Glendale, USA). The plate was incubated at 37 °C for 1 h and then washed in the same manner as the plates coated with the synthetic antigens and sLPS. The OPS iELISA was conducted as described for *B. abortus* sLPS antigen iELISA.

For the production of data shown in Figures 1 and 2, Appendix A, all the test results are expressed as a percentage of the signal generated from a common positive control serum applied to a control antigen (2,3,2-Tetra **(6)**) that was coated to all the ELISA test plates. For the production of data from the experimentally infected cattle the test results are expressed as a percentage of the signal generated from a common positive control serum applied to the individual antigen in question (Figures 3 and 4 and Appendix A). This same method was used for the generation of data from field samples, summarised in Table 2 and shown in Figure 5.

The Complement Fixation Test (CFT), Serum Agglutination Test (SAT), Fluorescent Polarisation Assay (FPA) and cELISA results for the serum samples from the experimentally infected cattle are as performed and presented previously [28].

### 2.6. Data Analysis

The calculation of the Area Under the receiver operator Curve (AUC), was performed using Graphpad Prism 7 software. The AUC metric was used to quantify the ability of the methods to differentiate between the populations being tested. Higher AUC values demonstrate that tests are better at discriminating between samples from two populations being tested. For example, a test that perfectly discriminates between populations (i.e., there is no overlap between the samples from the two different populations) has an AUC of 1, whereas a test that is completely random has an AUC of 0.5. In this study the two population types are samples from *Brucella* infected animals and samples from non-*Brucella* infected animals. Calculations for the planned significance testing for the differences between the AUC of the best synthetic antigen and the standard sLPS antigen *B. abortus* S99 [9] was performed using Microsoft Excel according to published methods [29].

Where a suitable positive/negative cut-off was identified, calculations of Diagnostic Sensitivity (DSn) and Diagnostic Specificity (DSp) were done in accordance with standard definitions [30]. DSn is the proportion of known infected (in this case infected with *Brucella*) reference animals that give positive results in the assay. DSp is the proportion of uninfected animals (in this case non-*Brucella* infected) that give negative results in the assay.

## 3. Results

### 3.1. Magnitude of Serological Response

The first study was an evaluation of the response of field sera to a panel of synthetic antigens including capped and non-capped equivalents. Figure 1 shows the iELISA results from 12 sera from *Brucella* infected animals (and average of four sera from *Brucella* non-infected animals) as tested against 11 different oligosaccharide BSA conjugate antigens including three (large circled) that are mannose capped. The results show that all the non-capped antigens are capable of differentiating between the samples from the infected and the non-infected animals. In general, there was a trend for the ELISA signals to reduce as the oligosaccharide antigens became smaller. However, the difference between the average results from the 2,2,2,2,2-Hexa **(9)** and the Mono **(1)** was only approximately 25%. While the exclusively 2,2-Tri **(5)** and 2,3-Tri **(3)** have similar properties, the 3,2-Tri **(4)** yields lower results. The 3,2-Tri **(4)** and 3-Di **(2)** also yield lower results on average than the Mono **(1)**.

The three lowest average results were from three capped antigens (Man-Mono **(11)**, Man-3,2-Tri **(12)**, and Tc-2,2,2,2,2,2-Hexa **(14)**). The other capped antigen, Man-2,2,2,2-Penta **(13)**, gave average results lower than a shorted uncapped equivalent **(5)**. Nearly all of the binding capability of the Mono **(1)** was lost when a mannose cap, Man-Mono **(11)**, was applied. For the longest capped oligosaccharides, most of the sera still contained antibodies that bound, but binding was overall much less than that of equivalent or even shorter uncapped antigens. One serum sample, infect 7, stood out as being capable of binding effectively to all the antigens, capped or not.

A similar study was conducted using capped and uncapped oligosaccharide variants but this time using an O-methyl group as the cap (conjugation to BSA was done using DSG as opposed to squarate as this was necessary for another study for which these antigens were originally prepared). These results are shown in Figure 2 and as with Figure 1 capping the antigen had a large negative influence on signal.

Where the same oligosaccharides have been used but with different linkers (squarate or DSG), the average signal is lower for the DSG conjugated antigens. For example, capped hexasaccharides **(14)** and **(21)**, 2,3,2-Tetrasaccharides **(4)** and **(17)**, 2,3-trisaccharides **(3)** and **(16),** 3-Disaccharides **(2)** and **(15).** This tallies with a lower number of oligosaccharides conjugated per BSA using DSG (7.3 to 9.5) compared to squarate (10–20).

### 3.2. Quantification of Impact of OPS Tip Epitope on Antibody Binding

The impact of the loss of the tip epitope was quantified by testing a dilution series of a subset of the samples (Appendix A). These results showed a 4-fold reduction in antibody binding between the capped pentasaccharide (Man-2,2,2,2-Penta **(13)**) and non-capped hexasaccharide (2,2,2,2,2-Hexa **(9)**), an approximately 8 fold reduction when the 3,2-Tri **(4)** is capped (Man-3,2-Tri **(12)**), an approximate 16 fold loss of titre when the disaccharide 3-Di-dsg **(15)** is capped (2Me-3-Di-dsg **(****18)**) and an approximate 64 fold reduction when the Mono **(1)** is capped (Man-Mono **(11)**). The data also shows that the 2,2,2,2,2-Hexa **(9)** has between a 4 and 8 fold higher titre than the Mono **(1)**.

The specific impact on the presence or absence of the tip on the binding of a MAb known to have specificity to linear epitope was evaluated using BM40. The results (Appendix A) confirm the M specific properties of this MAb, evident by the absence of binding to 2,2,2,2,2-Hexa **(9)** but high binding against oligosaccharides containing an α1,3 link. They also show that capping the 3,2-Tri **(4)** with mannose (Man-3,2-Tri **(12)**) does not decrease binding, but actually increased it to levels approximately equal to the 2,3,2-Tetra **(6)**.

A key observation is the impact of the antibody response to the tip epitope. This was evaluated using serum derived from an immunisation experiment conducted using *B. abortus* OPS. This OPS could not induce antibodies to the tip as this was eliminated by chemical modification [14]. When tested with the 2,2-Tri **(5)** (Appendix A), all the sera had titres less than 1/100. The same samples tested with the 2,2,2,2,2-Hexa **(9)** gave an average titre of 1/2900. When the same two antigens were tested against sera from cattle experimentally infected with *B. abortus*, where the OPS naturally presents with the tip, the difference in the average titres for the two antigens was only two-fold (Appendix A).

### 3.3. Diagnostic Impact of OPS Tip Epitope: Experimental Infection

To investigate the properties of the tip epitope further a selection of native, modified native and synthetic antigens were tested against serum from experimentally infected cattle. A representative selection of this data is shown in Figure 3 (*B. abortus*) and Figure 4 (*Y. enterocolitica* O:9). The data is presented as the average of the results from each of the four individual animals in each infection group per antigen shown over the time course of the infection period. The results for each antigen are shown as a separate line, expressed as a percentage of a common control serum as applied for each of the antigens. Therefore, unlike for the results for serum in Figure 1 and Figure 2, the results do not show the magnitude of the response. However, they do show antigen specific differences between the two infection types and antigen specific differences in the magnitude of the response over time.

The results from the *B. abortus* infected animals (Figure 3) shows the profiles for each of the six antigens (two mOPS antigens, the *B. abortus* S99 sLPS, the 2,2,2,2,2-Hexa **(9)**, 2,2-Tri **(5)** and Mono **(1)**) are very similar. The response increases from week 3 to 7 and remains broadly steady for the following two time points, weeks 16 and 24, before a small reduction at week 53. In contrast, sera from the *Y. enterocolitica* O:9 infected animals respond differently to the antigens (Figure 4). At one extreme, results for the *Y. enterocolitica* mOPS increases from week 3 to week 7 and then slowly decrease, but still end higher at week 53 than at week 3. At the other extreme the results for the 2-2-Tri **(5)**—essentially a predominantly tip type epitope—start high but fall sharply before levelling off. Results from the sLPS and 2,2,2,2,2-Hexa **(9)** antigens, both of which have tip and linear epitopes, show a similar but shallower decline. The *Brucella* mOPS antigen, only possessing linear epitopes, has a similar profile to the *Y. enterocolitica* O:9 mOPS, but with less of a pronounced increase across weeks 7, 16 and 24.

Appendix A show the individual sample results for the post infection sera from the *B. abortus* and *Y. enterocolitica* experimentally infected cattle as tested against 17 D-Rha4NFo-based antigens of variable source (native [*B. abortus* S99, *B. melitensis* 16M, *B. suis* biovar 2 Thomsen, *Y. enterocolitica* O:9] vs. synthetic), length, linkage, conjugation (direct to ELISA plate surface, natural [sLPS], BSA) and linker type (squarate, DSG). The mOPS antigens are conjugated to the ELISA plates via the non-reducing end so have no terminal structure available for antibody binding. Data for the capped oligosaccharides was not included in these figures as they responded poorly to the positive control, however the data is summarised in Table 2.

Appendix A shows the results for week 3 which demonstrate that for most antigens the response against sera raised by the two different infections is very similar. Notable exceptions where the response to sera from *Brucella* infected cattle was substantially higher on average, were the 3-Di **(2)**, Mono **(1)**, 2,3,2-Tetra-dsg **(17)** and 3-Di-dsg **(15)**.

Appendix A shows that week 7 post infection results for the sera from the *Brucella* infected animals have increased substantially against almost all antigens, results for the remainder (2,3-Tri **(3)**, 3,2-Tri **(4)**, Man-3,2-Tri **(12)** and 2Me-3-Di-dsg **(18)**) have remained at similar levels as week 3. In contrast the results from the *Y. enterocolitica* O:9 infected animals have fallen, with the exception of the mOPS antigens which have either risen or stayed the same. The week 16 results, Appendix A, show that results for the *Brucella* infected animals are still rising against most of the antigens. The results for the *Y. enterocolitica* O:9 infected animals continue to fall for most antigens but not for the mOPS and there remain clear signals for the 2,2,2,2,2-Hexa **(9)**, 2,3,2,2,2-Hexa **(8)** and 2,2-Tri antigens **(5)**.

The week 24 data, Appendix A, shows a fall in the response against antigens for both infection groups compared to the week 16 data. However, the mOPS results for the *Y. enterocolitica* O:9 infected animals are similar to the values at week 3 and there is also a clear response to the 2,2,2,2,2-Hexa **(9)** and 2,2-Tri **(5)** antigens but these have fallen since week 3. In contrast, the reaction to these two antigens from the sera from the *Brucella* infected animals has not decreased since week 3 (peaking at week 16).

The results for week 53, Appendix A, show a fall in the response of the serum to nearly all the antigens for both infection groups. However, the mOPS and sLPS results from the *Y. enterocolitica* O:9 infected animals do not appear close to reaching background levels, in particular the results from the mOPS antigens have hardly decreased in nearly a year. Also notable are the continually strong signals for the sera from the *Brucella* infected animals against the Mono **(1)** and 2,3,2-Tetra-dsg **(17)** and 3-Di-dsg **(15)** antigens even though the signals from the samples from the *Y. enterocolitica* O:9 infected animals against these antigens have almost completely disappeared.

Appendix A present the same data as shown in Appendix A, but each panel shows the results for one antigen type against time.

Table 2 columns 2 and 3 show summary statistics for each of the antigens from the testing done on the sera from the *Brucella* and *Y. enterocolitica* O:9 experimentally infected cattle. The highest specificity values are for the 3-Di **(2)**, 3,2-Tri **(4)**, 3-Di-dsg **(15)** and 2,3,2-Tetra-dsg **(17)** antigens. The lowest values were for the mOPS antigens extracted from *B. suis* biovar 2 and *Y. enterocolitica* O:9, the *B. abortus* sLPS antigen and the Man-3,2-Tri **(12)** antigen. The specificity results for some of the classical and contemporary tests, shown at the bottom of the table, were also poor. The highest AUC results were for the 3-Di-dsg **(15)** and 2,3,2-Tetra-dsg **(17)** > Mono **(1)** > 3-Di **(2)** > 3,2-Tri **(4)** > 2,2-Tri **(5)** > 2,3,2,2,2-Hexa **(8)**. The lowest AUC results were for the Man-3,2-Tri **(12)** < SAT < *Y. enterocolitica* O:9 mOPS < FPA < CFT.

### 3.4. Diagnostic Impact of OPS Tip Epitope: Field Samples

The superior antigens were tested against field sera. This included the 3-Di-dsg **(15)** and 2,3,2-Tetra-dsg **(****17)** antigens but they were rapidly screened out owing to poor sensitivity and specificity compared to their squarate conjugated counterparts (data not shown). The diagnostic capability of 11 of the antigens, as shown in Table 2 column 4, was further evaluated using a panel of sera comprising samples from 21 *Brucella abortus* infected cattle and FPSR samples from 16 cattle. The AUC for differentiation between these two sample populations was calculated for each antigen as shown in column 4. Some of these antigens were taken forward for additional testing on a total panel size of samples from 42 *Brucella abortus* infected cattle and FPSR samples from 39 cattle. The final ranking of the AUC values was 2,2-Tri **(****5)** > 3-Di **(****2)** > *B. abortus* S99 mOPS > 2,3,2-Tetra **(****6)** > *B. abortus* S99 sLPS > *B. melitensis* 16M mOPS > *Y. enterocolitica* O:9 mOPS. The difference between the antigen with highest AUC (2,2-Tri **(****5)**) and the standard antigen as used in iELISA (*B. abortus* S99 sLPS) was tested and shown to be highly significant (*p* = 0.00000002).

Figure 5 shows a scatterplot of the individual results from the *B. abortus* S99 sLPS and 2,2-Tri **(5)** antigens applied to the 42 samples from *Brucella* infected cattle and the 39 FPSR samples. It also shows results for 240 sera from randomly sampled GB cattle (brucellosis free). The plot shows that for the 2,2-Tri **(5)** 27 of 42 of the results from the sera from the *Brucella* infected animals exceed the highest result for the FPSR samples. For the sLPS antigen the highest result is actually from an FPSR sample. At the opposite end of the scale the lowest sLPS antigen result for the samples from the *Brucella* infected animals exceed the results for 6 FPSR samples, whereas for the 2,2-Tri **(5)** antigen this was the case for only 2 FPSR samples. The 2,2-Tri **(5)** antigen was more specific against the samples from the random non-infected animals as specificity and sensitivity were both 100%. In comparison the sLPS antigen, with a cut-off set to give 100% sensitivity, gave two false positive results (DSp = 99.17).

## 4. Discussion

The aim of this study was to identify the impact of the terminal structure on the detection of antibodies reactive to the *Brucella* OPS. A key part of this investigation was to test the serodiagnostic properties of different antigen structures. The sLPS from *B. abortus* S99 was included in the study as this is the reference antigen cited in the OIE Manual of Diagnostic Tests and Vaccines for production of iELISA and this is readily adsorbed to the polystyrene surface via the lipid-A. The oligosaccharides also adsorbed to the ELISA plate surface, in this case via the conjugated BSA. OPS was conjugated to ELISA plate surfaces via a chemical reaction targeting the terminal D-Rha4NFo [14] thus eliminating the putative tip epitope and leaving the linear epitopes intact.

The initial results demonstrated that the Mono **(1)** had a high potency, remarkable for such a small antigen. Indeed, in some cases it was more reactive than the longer oligosaccharides, possibly because it represents the tip antigen with no α1,2 or α1,3 linkage restriction. The second major finding was that capping the oligosaccharides with mannose, thus eliminating the putative tip epitope, had a very large and negative impact on antibody binding. The results clearly show that the tip of the OPS is a specific epitope and that antibodies of tip specificity constitute a greater proportion of those that bind the shorter oligosaccharides.

To counter the possibility that the mannose was having an active rather than passive inhibitory effect, oligosaccharides where the terminal unit was modified with an O-methyl were also tested. This is the type of cap that naturally occurs in the *V. cholera* O1 Ogawa serotype [19] and blocks the anti-tip antibodies induced by the Inaba serotype–which has identical OPS structure other than absence of the O-methyl group of Ogawa. In this study this O-methyl cap had a very similar inhibitory effect as the mannose addition.

Although the average signals for the 12 sera shown in Figure 1 and Figure 2 are lower for DSG conjugates for variants of the same oligosaccharide type, the averages are not vastly dissimilar considering the difference in oligosaccharide loading (squarate linkers have a stepwise reactivity which allows for more efficient conjugation reactions [31]). Despite the lower average, the range of the response is greater with the DSG conjugates. This may be a consequence of the bivalent nature of antibody binding. Conjugates with greater oligo loading may better facilitate bivalent antibody binding enabling antibodies with lower affinity to be better retained during ELISA and contribute to signal generation. Conjugates with lower oligo loading may be harder targets for antibodies to engage both paratopes [32] yet antibodies with higher affinity may effectively engage with a single paratope and be detected by ELISA. Notably, OPS tip-specific antibodies have been reported as having higher affinity than their anti-linear counterparts [21]. In this manner, for sera containing antibodies with higher binding affinity, conjugates with lower oligo loading may actually bind more antibodies and generate higher signals in ELISA.

The data from the BM40 binding study reinforced the body of evidence suggesting that removal of the tip epitope by capping with mannose or O-methyl was not actively inhibitory or repellent. This MAb bound very effectively to the capped oligosaccharides. Not only that but the addition of the mannose to the 3,2-Tri **(4)** to generate antigen Cman-3,2, Tri **(12)** actually enhanced the binding of this MAb thus supporting the selection of a mannose cap for this study.

The comparison of the 2,2-Tri and **(5)** and 2,2,2,2,2-Hexa **(9)** titres from the sera from the immunised mice (Appendix A) and experimentally infected cattle (Appendix A) suggests that most of the antibodies in sera from infected animals that bind to the 2,2-Tri **(5)** have some specificity to the tip. The sensitivity of the 2,2-Tri **(5)** against the sera from infected animals and lack of sensitivity against the sera from the mOPS immunised animals suggests that this antigen would be an effective DIVA diagnostic if used in combination with a mOPS protein conjugate immunogen [14].

The dependence on anti-tip antibodies to generate the strong signal for the 2,2-Tri **(5)** and the superior signal it generates compared to either the 2,3,2-Tetra **(6)** or the 2,3-Tri **(4)** does not fit comfortably with the most recent structural model of the A-dominant OPS [7]. In this, most of the OPS polymers possess a structure equivalent to the 2,3,2-Tetra **(****6)** at the non-reducing end. The degree to which the 2,2-Tri **(5)** bound antibodies from *Brucella* infected animals was therefore unexpected and it may be that the OPS polymers that are terminated with an additional α1,2 linked D-Rha4NFo are somehow more influential. It was also anticipated that this antigen may provide a stronger response against sera raised against the exclusively α1,2 linked OPS of *Y. enterocolitica* O:9.

The results from serum from the cattle experimentally infected with *B. abortus* and *Y. enterocolitica* O:9 showed a great deal of cross reaction but also that, in general, the serological results due to *Y. enterocolitica* O:9 infection fell faster than those due to *Brucella* infection. They also showed that the loss of sensitivity that occurs when short oligosaccharides with α1,3 links are capped overshadows any possible improvement in diagnostic specificity that may be gained from losing a non-*Brucella* specific structure. Thus, one initial point of interest within this study, that eliminating the common tip epitope would yield antigens with greater diagnostic specificity, became moot.

Most of the synthetic antigens showed greater specificity with this serum panel than the natural sLPS and the OPS antigens, apart from *B. melitensis*, some of the shorter oligosaccharides containing α1,3 linkages having particularly high values. Analysis using the AUC Figures showed a closer and slightly different picture. Nonetheless, many of the synthetic antigens still exceeded the AUC values of all the natural antigens.

Whereas the data from the α1,3 containing antigens was somewhat predictable based on prior evidence [13], there were several less anticipated findings from this study. Some antigens containing structures comprising exactly those found in the *Y. enterocolitica* O:9 OPS, with no *Brucella* unique components, gave DSp and AUC values better than expected. These were, in descending order of DSp and AUC, the Mono **(1)**, the 2,2-Tri **(5)** and 2,2,2,2,2-Hexa **(9)** antigens. Each gave DSp and AUC values higher than all native antigens; the sLPS and modified OPS. The ranking of these antigens suggests that the terminal D-Rha4NFo not only facilitates sensitive antibody detection but does so in a relatively specific manner and that addition of further α1,2-linked D-Rha4NFo units decreases specificity. This further suggests that epitopes comprising linear α1,2 D-Rha4NFo sequences are less specific to *Brucella* infection, compared to infection with *Y. enterocolitica* O:9. However, the tip epitope only appears to be more specific after the initial induction of antibodies following *Y. enterocolitica* O:9 infection (Figure 4). Although the results from the *Y. enterocolitica* O:9 samples generally fall over time for all antigens, the fall-off is quicker with the shorter oligosaccharide antigens, slower with the longer antigens and polysaccharides and with the mOPS antigens even initially increasing.

It was also notable that the DSG conjugates, 3-Di-dsg **(15)** and 2,3,2-Tetra-dsg **(17)** were more specific than their squarate counterparts—indeed any other antigen. This may be an indication that the sera from the animals experimentally infected with *Brucella* contained particularly high affinity anti-tip antibodies. If so, this may also explain why these antigens, and the Mono **(1),** were not sufficiently sensitive when subsequently applied to field samples.

The mOPS antigens did show a difference in results between the two infection types even if the samples from the *Y. enterocolitica* O:9 infected cattle never approached the baseline. The relatively poor specificity of the sLPS, even compared to mOPS, appears due to having both tip and long linear α1,2 D-Rha4NFo sequences. Consequently, recent infection with *Y. enterocolitica* O:9 leads to high serological results and no rapid decline subsequently. This is supported by the longitudinal profile of the 2,2,2,2,2-Hexa **(9)** antigen results that are very similar to that of the sLPS, the former having both the tip epitope and the longest linear α1,2 sequence of the synthetic antigens.

Within the mOPS antigen types there was evidence that antigens with α1,3 links were more specific. Also notable was the difference between the results for the two exclusively α1,2 linked mOPS antigens, *B. suis* bv 2 and *Y. enterocolitica* O:9 with the latter having inferior specificity. This could be due to differences in residual core sugars left at the reducing end of the OPS after mild acid hydrolysis [33]. Antibodies raised to the residual *Y. enterocolitica* O:9 core, or potentially epitopes that span the OPS and available core regions [34,35] may be responsible for some of the relatively elevated serological results from the sera from the cattle experimentally infected with *Y. enterocolitica* O:9 against the cognate antigen.

The results from the experimental infection suggest that during *Brucella* infection antibodies are raised against tip and linear epitopes in a similar proportion throughout infection. In contrast, during infection with *Y. enterocolitica* O:9 anti-tip antibodies are induced at high levels initially and then fall rapidly whereas antibodies against long linear α1,2 epitopes are sustained. The individual OPS structures of the infecting bacteria are very similar and would not initially appear responsible for the observed differences in antibody profile. However, it is known that *Y. enterocolitica* adapts its surface according to temperature [36]. At 37 °C it has also been reported that there is a reduction, or even loss, of OPS on the cell surface [37,38,39]. Although it must also be noted that not all studies support this [40]. By reducing what may be high density OPS packing [41] the cell surface of *Y. enterocolitica* O:9 may switch from one that is rich in tip epitopes with linear epitopes more sequestered to one in which those linear epitopes are very exposed and the tips less prominent. This potential alteration in OPS epitope presentation and availability may impact upon the development of the antibody repertoire and explain the data observed here.

Some of the more interesting antigens were evaluated with field FPSR samples and those from cattle confirmed as infected with *Brucella*. An initial population (where sera was available in higher volume) was evaluated with 10 different antigens and for the larger serum population the antigen number was reduced to seven. An equal mix of two antigens (2,2-Tri **(5)** and 2,3,2-Tetra **(6)**) was evaluated to test the feasibility of using an antigen combination. The *B. abortus* sLPS was also used as the reference antigen.

The results from the full set of field FPSRs and *Brucella* infected cattle identified the 2,2-Tri **(5)** as the superior antigen in this analysis. The results for the 3-Di **(2)** improved whereas results for the 2,3,2-Tetra **(6)** worsened with the addition of more samples to the analysis. All three of these antigens outperformed all the native antigens. Of these antigens the *Y. enterocolitica* O:9 mOPS antigen had particularly low AUC, much lower than that for the *B. abortus* mOPS. These results support the strong findings for the 2,2-Tri **(5)** and 3-Di **(2)** antigens against the sera from the experimentally infected cattle. Of note, the AUC value for the *B. melitensis* 16M mOPS was worse than expected. It was not possible to identify the aetiological agent of the field positive FPSRs so it is possible that agents other than *Y. enterocolitica* O:9 may be responsible [42].

The scatter plot in Figure 5 shows that the 2,2-Tri **(5)** is a more discriminatory antigen than the *B. abortus* sLPS and this difference is highly statistically significant. Although the 2,2-Tri **(5)** had superior AUC, the sLPS antigen did exclude more FPSR samples if the assay cut-off were placed at the maximum level that provides 100% DSn. The results from testing randomly sampled sera from non-*Brucella* infected cattle and selecting for 100% sensitivity gave 100% and 99.2% specificity with the 2,2-Tri **(5)** and sLPS respectively.

## 5. Conclusions

In conclusion, the results from this study show that the tip of the *Brucella* OPS is a specific antibody epitope. Although this is a common feature to *Brucella* and *Y. enterocolitica* O:9 OPS, this epitope is not only surprisingly sensitive, it is also surprisingly specific. Attempts to develop effective serodiagnostic assays for infections with smooth strains of *Brucella* with antigens not containing OPS have failed due to insufficient sensitivity. Not a single non-OPS-based alternative is available commercially or cited in the OIE manual [9]. Whilst OPS-based antigens remain vulnerable to cross reactions due to infection with bacteria possessing OPS structures similar to that of *Brucella*, this study has shown that cross reactions are reduced through the use of short synthetic antigens. This study shows that the 2,2-Tri **(5)** is the most effective antigen for the serodiagnosis of brucellosis in cattle. It combines excellent sensitivity and specificity and is significantly superior to the current gold standard sLPS antigen. Given that this is an exclusively α1,2 linked antigen this approach may also be appropriate for swine infected with *B. suis* biovar 2. It also appears that the 2,2-Tri **(5)** antigen would be DIVA compatible with vaccines developed with mOPS immunogens [14]. Synthetic antigens can be made to a high quality and purity with strict quality control using NMR and Mass Spectrometry techniques. They are also affordable, safe and eliminate the need to propagate large quantities of dangerous pathogens.

## Figures and Tables

**Figure 1 microorganisms-10-00708-f001:**
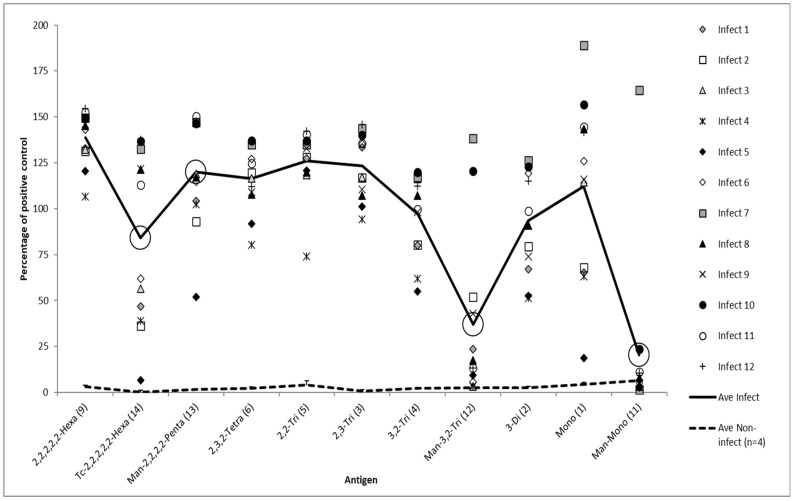
A line and dot plot showing synthetic oligosaccharide antigen structure (all squarate conjugates) on the *x*-axis and the cattle serum sample result, as a percentage of a common positive control sera against the 2,3,2-Tetra **(6)** antigen on the *y*-axis. The individual results for 12 serum samples, each tested for all antigens, are shown as individual icons (Infect 1–12). Samples 1–6 are from GB infected cattle and 7–12 are from Turkish infected cattle. The solid black line shows the average results for these 12 serum samples from infected cattle tested with 11 antigens. The dotted black line shows the average result of 4 serum samples from non-infected cattle (individual results not shown). The 11 antigens have been ordered along the *x*-axis according to size, with larger antigens, hexasaccharides, at the left end of the *x*-axis and smaller antigens, monosaccharides, on the right. The data from the antigens that are capped (all are mannose caps) are highlighted by large circles that surround the average result for the antigen.

**Figure 2 microorganisms-10-00708-f002:**
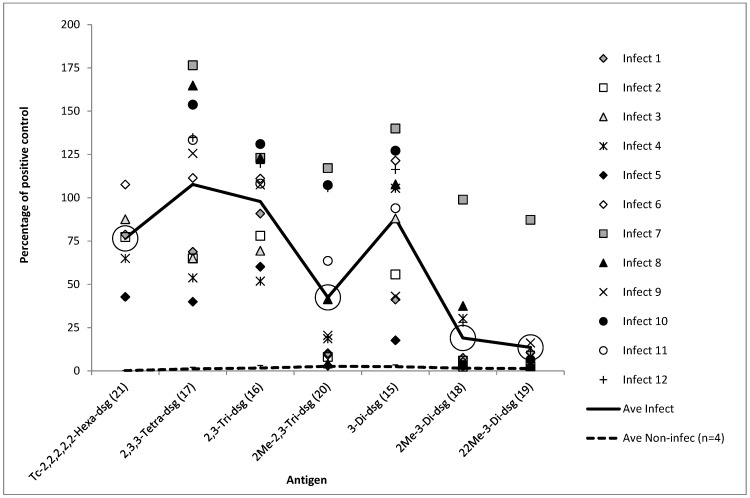
A line and dot plot showing antigen structure (all DSG conjugates) on the *x*-axis and the cattle serum sample result, as a percentage of the positive control sera against the 2,3,2-Tetra **(6)** antigen, on the *y*-axis. The individual results for 12 serum samples, tested with 7 DSG conjugated antigens, are shown as individual icons (Infect 1–12). Serum samples 1–6 are from GB infected cattle and samples 7–12 are from Turkish infected cattle. The solid black line shows the average results for these 12 serum samples from infected cattle. The dotted black line shows the average result of 4 samples from non-infected cattle (individual results not shown). The 7 antigens have been ordered according to size with larger antigens, terminally conjugated hexasaccharide, at the left end of the *x*-axis and smaller antigens, double capped monosaccharide, on the right. The data from the antigens that are capped (all are methyl caps) are highlighted by large circles that surround the average result for the antigen.

**Figure 3 microorganisms-10-00708-f003:**
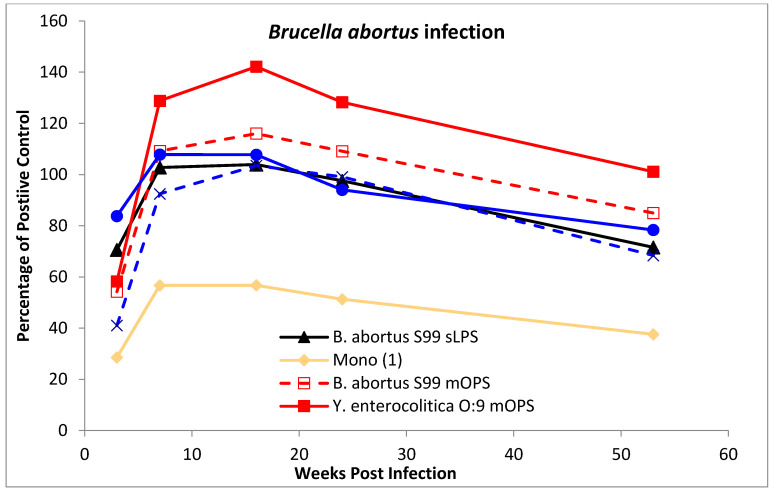
Average results from iELISA performed with sera collected from four cattle experimentally infected with *B. abortus* strain 544 and at five sampling dates (3, 7, 16, 24 and 53 weeks post infection) as show on the *x*-axis. The graph shows results against *B. abortus* S99 sLPS (solid black line), *B. abortus* mOPS (dashed red line), *Y. enterocolitica* O:9 mOPS (solid red line), 2,2,2,2,2-Hexa **(9)** (dashed blue line), 2,2-Tri **(5)** (solid blue line) and Mono **(1)** antigens (solid yellow line). The iELISA results (*y*-axis) are expressed as a percentage of a common positive control that was applied to each antigen type.

**Figure 4 microorganisms-10-00708-f004:**
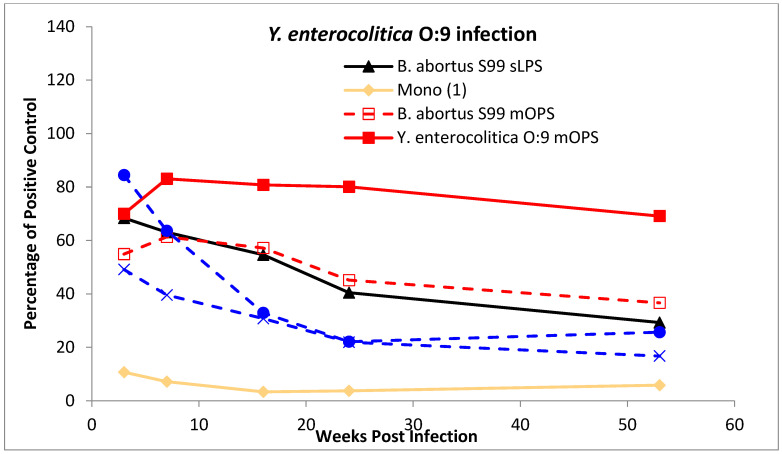
Average results from iELISA performed with sera collected from four cattle experimentally infected with *Y. enterocolitica* O:9 and at five sampling dates (3, 7, 16, 24 and 53 weeks post infection) as show on the *x*-axis. The sera were tested against *B. abortus* S99 sLPS (solid black line), *B. abortus* mOPS (dashed red line), *Y. enterocolitica* O:9 mOPS (solid red line), 2,2,2,2,2-Hexa **(9)** (dashed blue line), 2,2-Tri **(5)** (solid blue line) and Mono **(1)** (solid yellow line) antigens. The iELISA results (*y*-axis) are expressed as a percentage of a common positive control that was applied to each antigen type.

**Figure 5 microorganisms-10-00708-f005:**
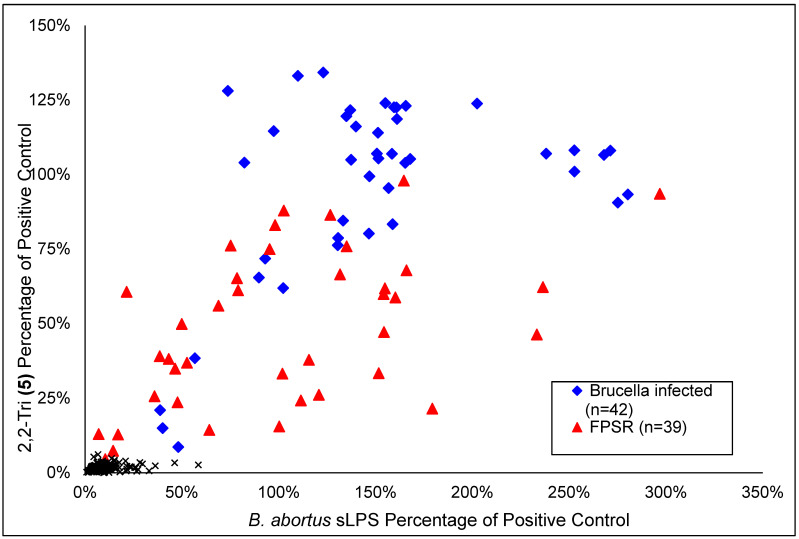
A scatter plot showing ELISA results using B*. abortus* S99 sLPS antigen on the *x*-axis and the 2,2-Tri **(5)** antigen on the *y*-axis. The results for three field sample populations are shown: sera from *B. abortus* infected cattle (*n* = 42), blue diamonds; FPSR cattle sera (*n* = 39), red triangles; and sera from randomly sampled cattle from GB an OBF country (*n* = 240), black crosses.

**Table 1 microorganisms-10-00708-t001:** Synthetic oligosaccharide BSA conjugates.

Antigen Number	Antigen Name	Oligosaccharide Structure	Linker Type	Oligo per BSA	Primary Reference
**(1)**	Mono	α-D-Rha4NFo	Sq	10-15	[14]
**(2)**	3-Di	α-D-Rha4NFo-(1→3)-α-D-Rha4NFo	Sq	15.2	[25]
**(3)**	2,3-Tri	α-D-Rha4NFo-(1→2)-α-D-Rha4NFo-(1→3)-α-D-Rha4NFo	Sq	15.9	[25]
**(4)**	3,2-Tri	α-D-Rha4NFo-(1→3)-α-D-Rha4NFo-(1→2)-α-D-Rha4NFo	Sq	15.7	[25]
**(5)**	2,2-Tri	α-D-Rha4NFo-(1→2)-α-D-Rha4NFo-(1→2)-α-D-Rha4NFo	Sq	16.2	[14]
**(6)**	2,3,2-Tetra	α-D-Rha4NFo-(1→2)-α-D-Rha4NFo-(1→3)-α-D-Rha4NFo-(1→2)-α-D-Rha4NFo	Sq	13.4	[25]
**(7)**	2,3,2,2-Penta	α-D-Rha4NFo-(1→2)-α-D-Rha4NFo-(1→3)-α-D-Rha4NFo-(1→2)-α-D-Rha4NFo-(1→2)- α-D-Rha4NFo	Sq	~16	[24]
**(8)**	2,3,2,2,2-Hexa	α-D-Rha4NFo-(1→2)-α-D-Rha4NFo-(1→3)-α-D-Rha4NFo-(1→2)-α-D-Rha4NFo-(1→2)- α-D-Rha4NFo-(1→2)-α-D-Rha4NFo	Sq	11.6	[25]
**(9)**	2,2,2,2,2-Hexa	α-D-Rha4NFo-(1→2)-α-D-Rha4NFo-(1→2)-α-D-Rha4NFo-(1→2)-α-D-Rha4NFo-(1→2)- α-D-Rha4NFo-(1→2)-α-D-Rha4NFo	Sq	13.8	[25]
**(10)**	2,2,2,3,2,2,2,2-Nona	α-D-Rha4NFo-(1→2)-α-D-Rha4NFo-(1→2)-α-D-Rha4NFo-(1→2)-α-D-Rha4NFo-(1→3)- α-D-Rha4NFo-(1→2)-α-D-Rha4NFo-(1→2)-α-D-Rha4NFo-(1→2)-α-D-Rha4NFo-(1→2)-α-D-Rha4NFo	Sq	~16	[24]
**(11)**	Man-Mono	α-D-Man-(1→2)-α-D-Rha4NFo	Sq	16	[26]
**(12)**	Man-3,2-Tri	α-D-Man-(1→2)-α-D-Rha4NFo-(1→3)-α-D-Rha4NFo-(1→2)-α-D-Rha4NFo	Sq	19.6	[26]
**(13)**	Man-2,2,2,2-Penta	α-D-Man-(1→2)-α-D-Rha4NFo-(1→2)-α-D-Rha4NFo-(1→2)-α-D-Rha4NFo-(1→2)-α-D-Rha4NFo-(1→2)- α-D-Rha4NFo	Sq	19	[26]
**(14)**	Tc-2,2,2,2,2-Hexa *	BSA-Sq-α-D-Rha-(1→2)-α-D-Rha4NFo-(1→2)-α-D-Rha4NFo-(1→2)-α-D-Rha4NFo-(1→2)-α-D-Rha4NFo-(1→2)- α-D-Rha4NFo-(1→2)-α-D-Rha4NFo	Sq	10.3	[14]
**(15)**	3-Di-dsg	α-D-Rha4NFo-(1→3)-α-D-Rha4NFo	DSG	Not done	Oligo as [14] conjugate not previously published
**(16)**	2,3-Tri-dsg	α-D-Rha4NFo-(1→2)-α-D-Rha4NFo-(1→3)-α-D-Rha4NFo	DSG	7.6	[26]
**(17)**	2,3,2-Tetra-dsg	α-D-Rha4NFo-(1→2)-α-D-Rha4NFo-(1→3)-α-D-Rha4NFo-(1→2)-α-D-Rha4NFo	DSG	7.3	[26]
**(18)**	2Me-3-Di-dsg	α-D-4-6-dideoxy-4-formamido-2-O-methyl-mannopyranosyl-(1→3)-α-D-Rha4NFo	DSG	9.2	[26]
**(19)**	2,3Me-3-Di-dsg	α-D-4-6-dideoxy-4-formamido-2-3-O-methyl-mannopyranosyl-(1→3)-α-D-Rha4NFo	DSG	8.7	[26]
**(20)**	2Me-2,3-Tr-dsg	α-D-4-6-dideoxy-4-formamido-2-O-methyl-mannopyranosyl-(1→2)-α-D-Rha4NFo-(1→3)-α-D-Rha4NFo	DSG	9.5	[26]
**(21)**	Tc-2,2,2,2,2-Hexa-dsg	BSA-DSG-α-D-Rha-(1→2)-α-D-Rha4NFo-(1→2)-α-D-Rha4NFo-(1→2)-α-D-Rha4NFo-(1→2)-α-D-Rha4NFo-(1→2)- α-D-Rha4NFo-(1→2)-α-D-Rha4NFo	DSG		Oligo as [14] conjugate not previously published

Conjugated to carrier (BSA) via anomeric carbon on reducing sugar via squarate linker unless stated (dsg suffix on name). * Conjugation is via terminal end.

**Table 2 microorganisms-10-00708-t002:** AUC and DSp for samples from experimentally infected cattle and for field samples.

Antigen	Specificity (Experimental)	AUC (Experimental)	AUC Field FPSRs (*n* = 16) vs. *Brucella* (*n* = 21)	AUC Field FPSRs (*n* = 39) vs. *Brucella* (*n* = 42)
Mono **(1)**	75	0.974	0.720	
3-Di **(2)**	85	0.957	0.807	0.834
2,3-Tri **(3)**	65	0.863		
3,2-Tri **(4)**	85	0.934		
2,2-Tri **(5)** *	65	0.916	0.908	0.888
2,3,2-Tetra **(6)**	55	0.882	0.795	0.730
2,2-Tri (5) + 2,3,2-Tetra **(6)**			0.839	
2,3,2,2-Penta **(7)**	40	0.867		
2,3,2,2,2-Hexa **(8)** *	40	0.916		
2,2,2,2,2-Hexa **(9)**	45	0.900	0.878	
2,2,2,3,2,2,2,2-Nona **(10)**	60	0.897		
Man-Mono **(11)**	Not done			
Man-3,2-Tri **(12)**	5	0.538		
Man-2,2,2,2-Penta **(13)**	45	0.870	0.795	
Tc-2,2,2,2,2-Hexa **(14)** *	40	0.816		
3-Di-dsg **(15)** *	85	0.976		
2,3-Tri-dsg **(16)**	Not done			
2,3,2-Tetra-dsg **(17)** *	85	0.976		
2Me-3-Di-dsg **(18)** *	55	0.892		
22Me-3-Di-dsg **(19)**	Not done			
2Me-2,3-Tri-dsg **(20)**	Not done			
Tc-2,2,2,2,2-Hexa-dsg **(21)**	Not done			
*B. abortus* S99 sLPS *	25	0.868	0.711	0.701
*B. abortus* S99 mOPS *	60	0.890	0.714	0.763
*B. melitensis* 16M mOPS *	70	0.897	0.607	0.650
*B. suis* (biovar 2) mOPS *	30	0.890		
*Y. enterocolitica* O:9 mOPS *	15	0.747	0.597	0.542
SAT (Sensitivity 85%, 30 I.U.s)	30 (50)	0.740		
CFT (Sensitivity 90%, 20 I.U.s)	0 (55)	0.848		
cELISA (*B. melitensis* 16M sLPS) (Sensitivity 90%)	15 (35)	0.853		
FPA (Sensitivity 90%)	45 (50)	0.843		

Specificity (generated by setting a positive/negative cut-off for each to be the highest value at which all samples from the *Brucella* experimentally infected animals were positive and then calculating the percentage of the 20 samples from the *Y. enterocolitica* O:9 experimentally infected animals that were negative at this cut-off). * for these antigens only 19 samples from the *Brucella* infected animals were tested—sample B2 from week 17 was insufficient in volume for all antigens. In this case the Sp calculation was conducted assuming that this sample did not set the positive/negative cut-off. For all other antigens the lowest *Brucella* result came from week 3 or week 53 (except for 2Me-3-Di-dsg **(19)** which was for week 16). AUC Experimental = AUC data generated from considering all samples from the *Brucella* experimentally infected animals as one population (*n* = 19, as the values for B2 week 7 was excluded because it was not universally present in all data sets) and all the samples from the *Y. enterocolitica* O:9 infected animals.

## Data Availability

Not applicable.

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
