# Peer review of "The Tip of Brucella O-Polysaccharide Is a Potent Epitope in Response to Brucellosis Infection and Enables Short Synthetic Antigens to Be Superior Diagnostic Reagents"

_microorganisms, 2022, doi:10.3390/microorganisms10040708_

Round 1

Reviewer 1 Report

I have only minor comments:

  1. Please check order of figures and their references, e.g. figure 5 is first figure in MS.
  2. There are many acronyms in the text. Sometimes full name is given later, sometimes not. Please ensure that a) acronym is defined after first appearance or b) provide a list of abbreviations and ensure completeness.

Author Response

1) Please check order of figures and their references, e.g. figure 5 is first figure in MS.

The authors agree with this comment. The figures and references were presented in the correct order for the original submission, but due to a journal formatting error the reviewers have been sent a version of the manuscript with figures in the wrong order. We will submit our final version with recommended changes and ensure that reviewers receive this version with the correct formatting.

2) There are many acronyms in the text. Sometimes full name is given later, sometimes not. Please ensure that a) acronym is defined after first appearance or b) provide a list of abbreviations and ensure completeness.

The authors will define the acronym after first appearance and provide a short list of key abbreviations.

Reviewer 2 Report

Dear Authors, 

Thank you for conducting research on important aspects of brucellosis diagnostics. 

The research article entitled "The tip of Brucella O-polysaccharide is a potent epitope in response to brucellosis infection and enables short synthetic antigens to be superior diagnostic reagents" highlights the identification and modification of OPS tip epitopes to increase the sensitivity and specificity in brucellosis diagnostic assays. 

The manuscript is well written, the methodology is well planned. However, the writing of the manuscript needs significant improvements. 

Additionally, the authors are requested to comment in detail or discuss the avoidance of cross-reaction owing to the results of the current research for other infections like Yersiniosis, etc that is not clearly explained in the manuscript. 

Another observation is the English grammatical mistakes and also poor sentence structuring. 

Thanks, 

Author Response

  • The manuscript is well written, the methodology is well planned. However, the writing of the manuscript needs significant improvements. 

The final manuscript was read by first and last authors. In addition, the manuscript was also read by senior and highly experienced authors from Imperial College, The University of Alberta and APHA, all of whom speak English as a first language. All considered the manuscript appropriate for publication. We are happy to try and address the reviewers concerns but need further guidance on where perceived failings lie. Please can the reviewer provide further guidance on areas of the manuscript that need significant writing improvements and the authors will make these changes.

  • Additionally, the authors are requested to comment in detail or discuss the avoidance of cross-reaction owing to the results of the current research for other infections like Yersiniosis, etc that is not clearly explained in the manuscript. 

The authors have revised the manuscript to include the following conclusion to address the reviewer’s comment:

Attempts to develop effective serodiagnostic assays for infections with smooth strains of Brucella with antigens not containing OPS have failed due to insufficient sensitivity. Not a single non-OPS based alternative is available commercially or cited in the OIE manual. Whilst OPS based antigens remain vulnerable to cross reactions due to infection with bacteria possessing OPS structures similar to that of Brucella, this study shows that these are reduced through the use of short synthetic antigens. 

  • Another observation is the English grammatical mistakes and also poor sentence structuring. 

Some errors have been spotted and corrected, but these are minimal in our view. As described above for point 1, we would appreciate some additional guidance from the reviewer so that we can better understand the necessity for this and how to remedy any faults. 

Reviewer 3 Report

The article is of a high professional level and relatively interesting, despite the fact that it is written in a way that is difficult to interpret. 

Some major considerations:

  1. It is not entirely clear how many total serum samples were tested (section 2.4). The authors used a very long description of the origin of the trials. I think it would be better to write it clearly and shorter. Moreover, the reference in this paragraph to Fig. 1 and Fig. 2 does not give any picture of the material under investigation because the figures are complicated and also do not give a simple answer to how many attempts we are dealing with.
  2. Cattle serum was sampled at various times post-infection (line 176). have these samples been marked as the same or as separate? This information cannot be deduced from the manuscript.
  3. Need to reformat the content layout. Figures are not cited in sequence

Minor comments:

line 58-64: the text is incorrectly formatted

line 129: please specify the incubation time

line 143: please remove the subscript when specifying the wavelength

line 166: it seems to me that more details should be provided on how the presence of B. abortus was confirmed using culture techniques

line 275-277, 287-290, Table 2 and other lines: The quoted references are given in square brackets, for compound designations please use other tags, in the current version it is misleading

Author Response

The article is of a high professional level and relatively interesting, despite the fact that it is written in a way that is difficult to interpret. 

Some major considerations:

  1. It is not entirely clear how many total serum samples were tested (section 2.4). The authors used a very long description of the origin of the trials. I think it would be better to write it clearly and shorter. Moreover, the reference in this paragraph to Fig. 1 and Fig. 2 does not give any picture of the material under investigation because the figures are complicated and also do not give a simple answer to how many attempts we are dealing with.

The authors will improve the description for methods in section 2.4, to clarify the number of serum samples and the provenance of the samples. We have tried to be as concise as possible with the description but are conscious of the need to be as exact as possible about provenance as this is key to understanding the significance of the results. We will also improve the figure 1 and 2 legends to better explain these results.

  1. Cattle serum was sampled at various times post-infection (line 176). have these samples been marked as the same or as separate? This information cannot be deduced from the manuscript.

The authors will amend this sentence to say, “Samples that were taken 3, 7, 16, 24 and 53 weeks post infection from each animal (n=8) at each time point were evaluated in this study with a panel of diagnostic antigens. The average results for iELISA with sera from the four cattle experimentally infected with B. abortus were evaluated with a panel of antigens are shown in Figure 3 and 4”.

  1. Need to reformat the content layout. Figures are not cited in sequence

The authors agree with this comment. The figures were presented in the correct sequence for the original submission, but due to a journal formatting error the reviewers have been sent a version of the manuscript with figures in the wrong order. We will submit our final version with recommended changes and ensure that reviewers receive this version with the correct formatting.

Minor comments:

  • line 58-64: the text is incorrectly formatted

This paragraph will be reformatted and the described structure will be labelled as scheme 1, with a legend to clarify these sentences.

  • line 129: please specify the incubation time

These sentences will be amended to state 3-5 days late-log phase. “The strains were grown on serum dextrose agar plates at 37ºC at 10% CO2 for 3-5 days until late-log phase and harvested into phosphate buffered saline (PBS). Y. enterocolitica serotype O:9 (biotype 2), from which OPS was extracted, was grown on nutrient agar plates at 25ºC, atmospheric CO2 levels for 3 days until late-log phase and then harvested into PBS.”

  • line 143: please remove the subscript when specifying the wavelength. Yes, the authors will amend this.

  • line 166: it seems to me that more details should be provided on how the presence of B. abortus was confirmed using culture techniques.

The authors will amend this line as follows: “Another 6 serum samples (7-12) were derived from cattle in Turkey and were serologically positive by RBT and CFT. These samples were from a herd where at least one animal had been confirmed as being infected with an A dominant strain of B. abortus (biovars 1 or 3) by culture from milk samples.

  • line 275-277, 287-290, Table 2 and other lines: The quoted references are given in square brackets, for compound designations please use other tags, in the current version it is misleading

The authors will amend the compound designations so that they are distinguishable from the references. References in this journal are usually denoted in blue font, so as to distinguish from the text, however there were journal formatting issues for the copy of the manuscript that the reviewers have received.

Round 2

Reviewer 3 Report

Interesting article. Congratulations!